# R-Tac: A Rounded Monochrome Vision-based Tactile Sensor

Wanlin Li[1*], Pei Lin[1,2*], Meng Wang[1], Chenxi Xiao[2],
Kaspar Althoefer[3], Yao Su[1†], Ziyuan Jiao[1†], Hangxin Liu[1†]

*Abstract*— Endowing the curved surfaces of rounded vision-based tactile sensors (VBTS) is essential for dexterous robotic manipulation, as they offer more sufficient contact with the environment. However, current rounded designs are constrained by a low sensing frequency (30–60 Hz) and the need for extensive calibration when adapting to new sensors due to the reliance on multi-channel captures, which hinders their performance in dynamic robotic tasks and large-scale deployment. In this work, we introduce R-Tac, a low-cost rounded VBTS engineered for high-resolution and high-speed perception. The key innovation is a monochrome vision-based sensing principle: utilizing a single-channel camera to capture the reflection properties of the compound rounded elastomer under monochromatic illumination. This single-channel sensing principle significantly reduces data volume and simplifies computational complexity, enabling 120 Hz tactile perception. R-Tac features an efficient calibration process, requiring only a few captures with a 3D-printed calibration setup. We demonstrate the advantages of R-Tac's design through a series of tasks (shown in Fig. 1) including object grasping, paper picking, in-hand object reorientation, and terrain exploration, showcasing its effectiveness for dexterous robotic hands.

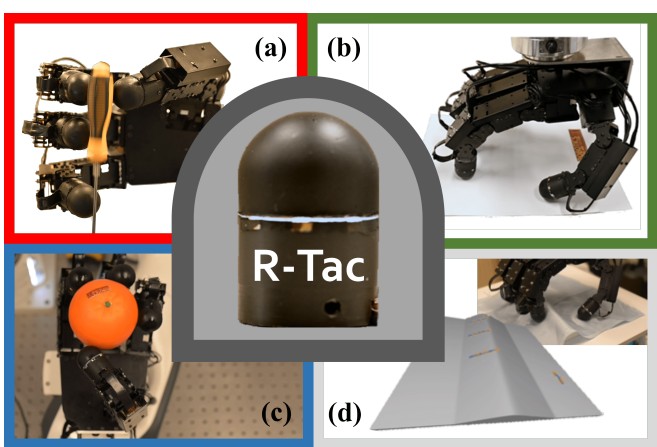

Fig. 1: **The R-Tac Tactile Sensor Designed For Fingertips of Dexterous Robotic Hands.** (a) Object grasping task. (b) Paper picking task. (c) In-hand object reorientation task. (d) Terrain exploration task.

## I. INTRODUCTION

Tactile sensors [1, 2] are essential for robotic manipulation, providing precise feedback on contact states, contact positions, and surface characteristics. In recent years, Vision-based Tactile Sensors (VBTSs) [3–14] have made significant progress, while most designs are flat, and their responseness remains limited (running under 100 Hz compared to e-skin [15]), which can cause missed detections and even motion blur during dynamic robotic tasks. This stems from their reliance on multi-channel RGB image data, which increases computational load and heat generation.

When deploying VBTSs on robotic multi-finger dexterous hand, factors such as calibration process and cost in large-scale deployments are also important. The complexity of multi-channel data requires individual and extensive calibration for each newly manufactured sensor, and this calibration necessitate specialized equipment (*e.g.* CNC machines). These pose significant challenges to the efficient large-scale deployment of VBTSs at present.

Our response to this overall challenge is the development of R-Tac, that leverages a *monochrome* vision-based sensing principle. The proposed rounded sensor uses a low-cost single-channel camera to capture the reflection properties of a coated semitransparent elastomer under white-lighting illumination, enabling *high-frequency* (120 Hz) *pixel-level curved surface reconstruction* and *simple calibration*. The comparison between R-Tac and current VBTS is shown in Tab. I. We also introduce an efficient, 3D-printed calibration setup that requires only a few captures. Therefore, R-Tac is a novel spherical tactile sensor designed with ease of fabrication, calibration, and scalable deployment. To demonstrate its utility, we integrate R-Tac into each fingertip of a fully actuated dexterous robotic hand, enabling real-time contact feedback during manipulation tasks.

## II. SENSOR DESIGN

The design criterial of R-Tac tactile sensor (Fig. 3) is guided by five key principles to ensure effective integration for robotic end-effectors:
- **Round shape:** The hemispherical design enables omnidirectional tactile perception.
- **High resolution:** High resolution enables accurate depth reconstruction and slip detection during picking-up.
- **Convenient to fabricate & low-cost:** The components of the tactile sensor are either off-the-shelf or easy to fabricate, with a cost of around $60.
- **Efficient calibration:** The monochrome sensing principle simplifies lighting control and reduces manual effort for calibration, making it particularly suitable for large-scale deployment on multi-fingered robotic hands.

* Wanlin Li and Pei Lin contributed equally to this work.
† Corresponding authors.
[1] State Key Laboratory of General Artificial Intelligence, Beijing Institute for General Artificial Intelligence (BIGAI). Emails: {liwanlin, linpei, wangmeng, suyao, jiaoziyuan, liuhx}@bigai.ai.
[2] School of Information Science and Technology, ShanghaiTech University. Email: xiaochx@shanghaitech.edu.cn.
[3] Centre for Advanced Robotics @ Queen Mary (ARQ), Queen Mary University of London. Email: k.althoefer@qmul.ac.uk.

TABLE I: **Comparison of the proposed R-Tac with the state-of-the-art curved VBTSs**

| Sensor | Working Principle | Camera | Dimension (mm) | Cost ($) | Frequency (Hz) | Configuration |
|---|---|---|---|---|---|---|
| TacTip [5] | Learning-based | Monocular RGB | $40 \times 40 \times 85$ | - | 90 | Bionic fingertip |
| RainbowSight [16] | Photometric Stereo | Monocular RGB | $28 \times 28 \times 50$ | - | 30 | Bionic fingertip |
| Omnitact [17] | Photometric Stereo | 5 Cameras | $30 \times 30 \times 33$ | 3200 | 30 | Bionic fingertip |
| GelTip [18] | Photometric Stereo | Monocular RGB | $30 \times 30 \times 100$ | - | 30 | Bionic finger |
| InSight [19] | Photometric Stereo | Monocular RGB | $40 \times 40 \times 70$ | - | 40 | Bionic finger |
| AllSight [20] | Photometric Stereo | Monocular RGB | $26 \times 28 \times 38$ | 30+ | 60 | Bionic fingertip |
| DenseTact [12] | Learning-based | Monocular RGB | $32 \times 32 \times 43$ | 80- | 30 | Bionic fingertip |
| DIGIT Pinki [13] | Photometric Stereo | Monocular RGB | $15 \times 15 \times 15$ | - | 30 | Bionic fingertip |
| GelStereo BioTip [21] | Binocular Stereo | Binocular RGB | $34 \times 28 \times 34$ | - | 60 | Bionic fingertip |
| DTact [6] | Darkness Mapping | Monocular RGB | $32.5 \times 25.5 \times 25.5$ | 15 | 90 | Non-planar |
| **R-Tac (Ours)** | Darkness Mapping | Monochrome | $30 \times 30 \times 43$ | 60 | 120 | Bionic fingertip |

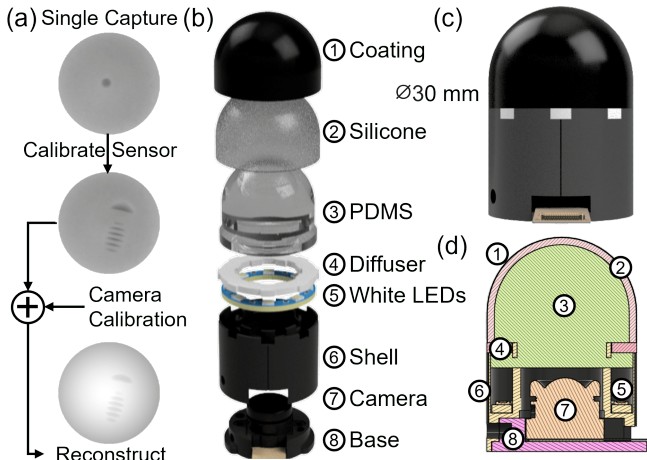

Fig. 2: **R-Tac Sensor Design and Calibration.** (a) illustrates the pipeline of depth reconstruction. (b) illustrates the exploded view of the sensor, detailing each component. (c) shows the dimensions of the sensor. (d) shows the schematic design.

- **Efficient data transmission:** The monochrome camera produces lightweight data per frame, facilitating high-speed data transmission between systems.

## III. SENSOR CALIBRATION

The uniform optical properties of the elastomer and illumination module (standard deviation as low as 6) enable the 3D geometry of the round shape sensor to be computed from single-channel pixel intensity in simply two steps using only 30 captures. First, given the known intrinsic parameters $K$, camera calibration ( Fig. 3) is performed using 29 captures in a 3D-printed indentation-based setup to estimate the extrinsic parameters of rotation matrix $A$ and translation vector $b$, as well as the sensor surface reference projection $D$. Next, the depth mapping function $M$ is calibrated by capturing a single image of a ball of known size pressed onto the sensor [6]. The mapping function from the pixel coordinates $(u, v)$ to the sensor coordinates $(x, y, z)$ can be expressed as:

$$\begin{bmatrix} x \\ y \\ z \end{bmatrix} = A^{-1} \left( (D(u,v) - M(I_\Delta(u,v)))K^{-1} \begin{bmatrix} u \\ v \\ 1 \end{bmatrix} - b \right), \quad (1)$$

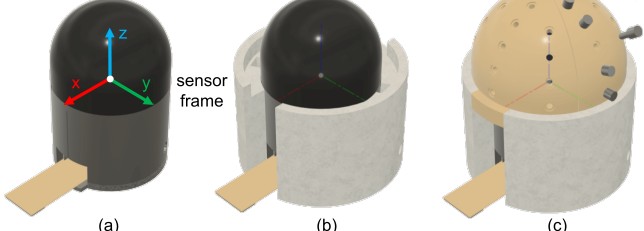

Fig. 3: **Camera Calibration.** We calibrate the intrinsic parameters and distortion using a calibration board. We 3D print a dome structure with predetermined holes, and by inserting pins at known coordinates, we are able to obtain pair-wise 3D and 2D points. The camera pose is then obtained using OpenCV's solvePnP function.

which transforms grayscale intensity images to a depth map expressed in the sensor coordinates. Moreover, R-Tac sensor is capable of detecting both deformation and slip events. We trained a lightweight neural network for slippage detection.

We quantify the reconstruction error by leveraging ground truth indentation information obtained from 3D-printed hemispherical shape indicators containing various testing indenters. We collected 215 testing configurations, each with paired sensor outputs and ground truth reprojection images. The sensor achieves a mean absolute error (L1 error) reconstruction loss of $0.35\ mm$, and a median loss of $0.28\ mm$, with 60% of reconstruction losses below $0.3\ mm$. In terms of computational speed, the depth mapping process takes less than $10\ ms$, ensuring real-time performance for robotic applications.

## IV. CONCLUSION

In this work, we present R-Tac, a round-shaped, low-cost monochrome vision-based tactile sensor that achieves pixel-level surface reconstruction at a high-frequency of 120 Hz. We have also developed an easy-to-deploy calibration method that relies solely on 3D-printed setups, requiring only a few captures to achieve robust performance. R-Tac is designed to be compatible with the fingertips of current dexterous robotic hands, such as the Allegro Hand and Leap Hand, enabling them to perform various manipulation and exploration tasks. To support further research and community development, the design of R-Tac is open-sourced.

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
