# OpenReview forum: "R-Tac: A Rounded Monochrome Vision-based Tactile Sensor"
_IEEE.org/IROS/2025/Workshop/Tactile_Sensing — IROS 2025 Workshop Tactile Sensing Poster_

### Official Review · Reviewer_RtNm · 2025-09-20
**Monochrome vision-based tactile sensor**

**Rating:** 6
**Confidence:** 4

**Review:**

This paper presents a vision-based tactile prototype that is solely based on a single monochrome camera that with the output frame rate of  120 Hz. Using the pixel-wise linear regression mapping, the deformation shape and depth is obtained.

Based on personal experience, I wonder whether using monochrome darkness to calculate deformation will compromise the accuracy compared to photometric stereo. The author is recommended to explain why it prevails the dominant photometric stereo method.

---

### Official Review · Reviewer_1JAT · 2025-09-20
**A Rounded Monochrome Vision-based Tactile Sensor**

**Rating:** 8
**Confidence:** 4

**Review:**

This is an interesting work that proposes a depth reconstruction method based on monochromatic illumination, and expands the current reconstruction scheme based on trichromatic illumination. The reviewer is very much looking forward to seeing the actual reconstruction results. Furthermore, the reviewer wants to know the specific model of the camera and is interested in the 120 Hz tactile perception.